# Integrated model for Food-Energy-Water (FEW) nexus to study global sustainability: The water compartments and water stress analysis

Neeraj Hanumante[1]☯, Yogendra Shastri[1]☯, Apoorva Nisal[2]‡, Urmila Diwekar📧[2,3]‡*, Heriberto Cabezas[4]‡

1 Department of Chemical Engineering, Indian Institute of Technology Bombay, Mumbai, Maharashtra, India, 2 Department of Industrial Engineering, University of Illinois, Chicago, IL, United States of America, 3 Vishwamitra Research Institute, Crystal Lake, IL, United States of America, 4 Research Institute for Applied Earth Sciences, University of Miskolc, Miskolc, Hungary

☯ These authors contributed equally to this work.
‡ AN, UD and HC also contributed equally to this work.
* urmila@vri-custom.org

**Data Availability Statement:** All relevant data are available on Zenodo: https://zenodo.org/record/6331602.

**Funding:** NH, YS, UD acknowledge the support from the Ministry of Human Resource

## Abstract

Analysis of global sustainability is incomplete without an examination of the FEW nexus. Here, we modify the Generalized Global Sustainability Model (GGSM) to incorporate the global water system and project water stress on the global and regional levels. Five key water-consuming sectors considered here are agricultural, municipal, energy, industry, and livestock. The regions are created based on the continents, namely, Africa, Asia, Europe, North America, Oceania, and South America. The sectoral water use intensities and geographical distribution of the water demand were parameterized using historical data. A more realistic and novel indicator is proposed to assess the water situation: net water stress. It considers the water whose utility can be harvested, within economic and technological considerations, rather than the total renewable water resources. Simulation results indicate that overall global water availability is adequate to support the rising water demand in the next century. However, regional heterogeneity of water availability leads to high water stress in Africa. Africa's maximum net water stress is 140%, so the water demand is expected to be more than total exploitable water resources. Africa might soon cross the 100% threshold/breakeven in 2022. For a population explosion scenario, the intensity of the water crisis for Africa and Asia is expected to rise further, and the maximum net water stress would reach 149% and 97%, respectively. The water use efficiency improvement for the agricultural sector, which reduces the water demand by 30%, could help to delay this crisis significantly.

## Introduction

Water is essential for human survival and critical for other spheres of human civilization, namely, agriculture, municipal, industrial, livestock, and energy production. Agriculture, responsible for food and feed production, is the largest consumer of water (50%-90% for

Development, Government of India, through the SPARC (project code: P1238). The research contribution by H. Cabezas was carried out in the GINOP-2.3.2-15-2016-00010 framework Development of enhanced engineering methods with the aim at utilization of subterranean energy resources" project at the Research Institute of Applied Earth Sciences of the University of Miskolc, the Széchenyi 2020 Plan, partially funded by the European Union, co-financed by the European Structural and Investment Funds. The funders had no role in study design, data collection and analysis, decision to publish, or preparation of the manuscript.

**Competing interests:** The authors have declared that no competing interests exist.

various countries) as compared to other anthropocentric consumption routes [1]. Energy is an important contributor to human development and well-being, and it is also a key input to agriculture [2]. With electricity-powered pumps, water can be drawn from deeper wells or transferred for long distances, highlighting the role of energy in improving water accessibility. However, energy production requires water, and some energy production technologies are highly water-intensive. Biopower and biofuels have been shown to have high water footprints (40–400 cu m per GJ electricity) [3]. Additionally, excessive water withdrawal due to cheap energy may further deplete groundwater resources [4]. Thus, ensuring adequate water availability for everyone is a complex multi-dimensional problem that cannot be addressed in isolation. Rather, it is a part of the food-energy-water (FEW) nexus.

The inter-dependencies of these three sectors are complex. The international community has understood the importance of a sustainable water supply. It can be seen from SDG6: clean water and sanitation for all, which has been incorporated in Agenda 2030 [5]. Scanlon *et al.* [6] highlighted the criticality of understanding the FEW nexus. Several researchers have studied the FEW nexus with different objectives. For example, D'Odorico *et al.* [7] closely examined the individual links between food-energy, energy-water, and food-water and identified the critical areas for action. However, this study did not capture the integrated nature of the FEW nexus. The scale of the FEW nexus is also an essential factor. However, Nie *et al.* [8] studied the optimization problem involving the FEW nexus on a small scale. Several research groups [9, 10] studied the FEW nexus at the regional scale, which again represents a limited spatial scale. Nevertheless, as global systems are large, inter-dimensional influences may take time to manifest. Hence, integrated cross-disciplinary models exploring the interactions between these various components have become increasingly important. Such models and their applications can improve our understanding of the complex relationships between these sectors and provide prescriptive recommendations using scenario analysis.

Development of such global models began with World3 by the Club of Rome [11]. The World3 model included subsystems such as food production, industry, population, pollution, and non-renewable resources. Some of the other notable examples of global models are GUMBO [12], Earth3 [13], and HANDY [14]. More recently, Nisal *et al.* [15] developed the Generalized Global Sustainability Model (GGSM). As compared to other global models, GGSM's ecological dimension is represented with features such as multiple trophic levels and intra-trophic level diversity. Moreover, the explicit integration between the economic and ecological dimensions is another strong point. Further, this model is both descriptive and prescriptive compared to other global models and can be used to derive techno-socio-ecological policies.

One fundamental limitation of the revised model was that it did not explicitly capture water as a resource. It was assumed to be a part of the generalized resource pool. As highlighted previously, water is a crucial resource and part of the FEW nexus. By its definition, sustainable water supply aims at consistent water availability over the foreseeable future for all of humanity. Based on this, an analysis of long-term global demand-supply dynamics of water for various sectors is necessary for understanding the criticality of the situation. Currently, prima facie, the available water is adequate to match human consumption on an aggregate level [16]. But, the situation changes when different sectors and regional perspectives come into the picture. Some existing studies are investigating the dynamics of water stress. However, these studies have a different focus; for example, a study by Konapala *et al.* [17] studied the influence of climate change on global water supply at a high resolution of 2.5. . . × 2.5. . .. On the other hand, Terrapon-Pfaff *et al.* [18] analyzed the water demand scenarios for a single sector, that is, the electricity sector. This motivates us to investigate the long-term regional demand-supply dynamics for various sectors and analyze the regional stress.

Water Stress Indicator, defined as the ratio of total water withdrawal to total renewable water, is traditionally used to quantify the severity of the water crisis [19, 20]. However, as identified by Gain *et al.* [21], this indicator considers only renewable water; that is, groundwater depletion is not included in the water use. Hence, developing an indicator that can capture the real water stress and be used for future projections becomes necessary. Increasing population is considered one of the major challenges for years to come; hence, it is prudent to investigate how an accelerated population growth influences the water demand-supply dynamics. Lastly, the SDG 6 measures are expected to improve the overall water management [22]. Long term efficacy of such measures needs to be examined.

The critical novel contribution of this work is the revision of GGSM to incorporate a global water network and scenario studies to propose policy alternatives. On a broader level, this analysis work tackles the following questions using the GGSM: with business-as-usual conditions, can we have a sustainable water supply for all? How would the situation change if there was a population explosion? Can the improvements under Sustainable Development Goal 6 help in preventing undesirable situations?

Following this introduction, the revision of GGSM to incorporate the global water system and the modelling of regional dynamics is addressed in the next section. The scenario planning to address the aforementioned questions is presented in Section Scenario planning. Section Results and discussion discusses the simulation results and their implications. The last section presents concluding remarks.

## Modelling global water system

The GGSM model developed by Nisal *et al.* [15] focussed on reparameterization and capturing the historical trends of the global population, carbon emissions, GDP, and $NO_x$. However, water-related compartments and flows were not modeled. As already mentioned, this limitation is addressed here.

### Scope and assumptions

The water system modeling in the GGSM model is carried out in two steps: first, the global water reservoirs and flows are established, and then second, the demands by different sectors are modeled. The critical assumptions are listed below:

1. Seawater is ignored.

2. Only exploitable water, that is, the total surface water and regular renewable groundwater, is considered.

3. Fossil groundwater, desalinated water, environmental water requirements, and flows are not considered.

4. Effect of climate change and extreme weather events are not modeled in the present effort, but they could be addressed using this paradigm in future work.

5. Economics of water supply-demand is not considered.

6. Historical sectoral water intensity trends are assumed to be valid in future and the possibility of a disruptive technology becoming available is ignored.

The scope of this work is limited to enhancement of the GGSM to capture the historical availability-demand dynamics on a global and regional level and then analyze the projections of these dynamics.

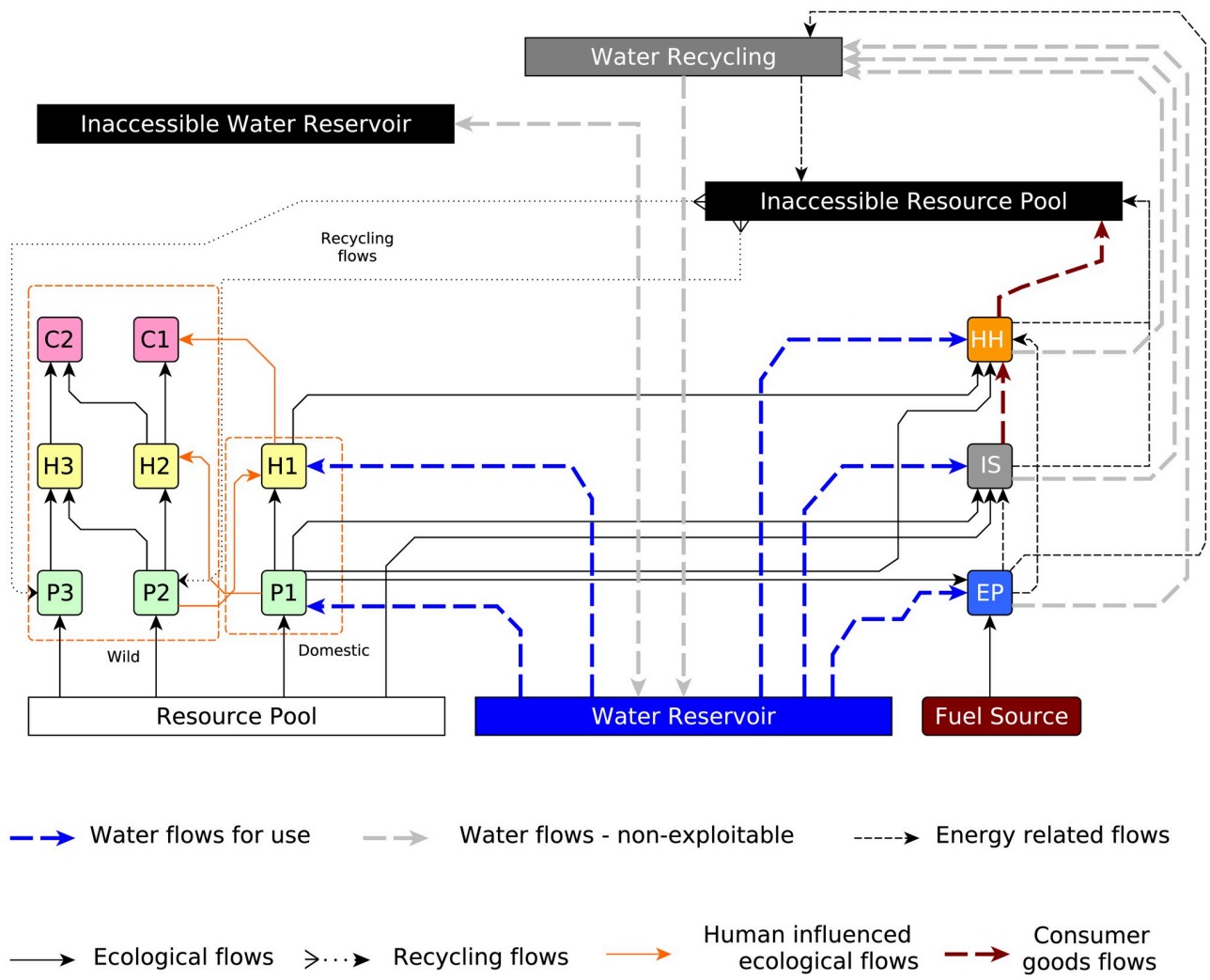

**Fig 1. The generalized global sustainability model (GGSM): The water sector in the GGSM model consists of three compartments, namely, water reservoir, inaccessible water reservoir, and water recycling.** Water withdrawal by various sectors from the Water Reservoir is shown by dashed blue lines. Grey dashed lines indicate non-consumption water flows, that is, industrial and municipal water sent for recycling and the direct water transfer between the three compartments.

## Model details

In Fig 1, the water reservoir represents the common pool of global exploitable water. Exploitable water is the aggregation of the total surface water, and renewable groundwater [23]. It represents the quantity of water that is consistently available and can be used with reasonable economic and technological investments, for example, lakes and wells. AUQASTAT defines it as follows:

> Exploitable water resources (also called manageable water resources or water development potential) are considered to be available for development, taking into consideration factors such as: the economic and environmental feasibility of storing floodwater behind dams, extracting groundwater, the physical possibility of storing water that naturally flows out to

the sea, and minimum flow requirements (navigation, environmental services, aquatic life, etc). Methods to assess exploitable water resources vary from country to country.

On the other hand, as the name suggests, an inaccessible water reservoir represents inaccessible water whose utility cannot be harvested owing to technological or economic limitations, for example, water resources far away from consumption centers. As water is a low-cost, high-volume commodity, under normal circumstances, the transportation of water from large distances is not economically feasible. Another example of inaccessible water is the surface run-offs in urban locations. Unless rainwater harvesting is extensively implemented, the utility of this water cannot be exploited. As the name suggests, the Water Recycling compartment represents the recycling of water outflows from the municipal, industrial, and energy sectors.

Nisal *et al.* [15] developed the GGSM with historical data and has projections up to 130 years into the future. Sectoral water intensity trends are used to obtain the water demands for multiple sectors. Sectoral water intensity represents the quantity of water withdrawn per unit of the desired GGSM variable. Agriculture and livestock are part of the ecological and economic systems and require water to survive. The water demand for agricultural and livestock sectors is computed considering this constraint as a function of P1 and H1 masses, respectively. On the other hand, the industrial and energy sectors do not need continuous water replenishment. These sectors withdraw water during the production process only. Hence, water demand for the industrial and energy sectors is computed as a function of IS and EP production. The municipal water demand is computed as a function of the human population.

Eq 1 shows the utility of sectoral water intensity $\Psi$ to obtain the total sectoral water demand $Ts$ for sector $s$ at timestep $i$ from GGSM variables $Y$. Here, $s$ is a sector in Agriculture (*P1*), Municipal (*HH*), Industry (*IS*), Energy (*EP*), or Livestock (*H1*).

$$Ts_s^i = Y_s^i \times \Psi_s \tag{1}$$

The region under consideration greatly influences the demand-supply dynamics of water. The countries are categorized into six groups to study this aspect based on the continent where they are located, namely, Africa, Asia, Europe, North America, Oceania, and South America.

Water flows from the water reservoir to *P1*, *H1*, *IS*, *EP*, and *HH* represent water consumption by agricultural, livestock, industrial, energy, and municipal sectors. The agricultural sector uses the water for irrigation purposes. It includes the surface water obtained from lakes, rivers, and canals; and the groundwater pumped from wells. For the livestock sector, the water is mainly used to survive and maintain hygiene. Industrial and energy sectors use the water for the production process. For all these sectors, though water is essential for its functionality, it is assumed that water does not contribute to the mass of that compartment; that is, embedded water is not considered.

As exact details of water demand and supply distributions for various sectors are not available, other representative variables are used to obtain each sector's regional distribution of the water variables.

The total value of the representative variable $v$ for region $r$ represented by $X$ can be obtained using Eq 2a. Its distribution, that is, contribution of region $r$ to the global value of sector $s$ is represented by $G$. Here, $v$ is a representative variable in Agricultural area, GDP, Meat production, or Population; $c$ is a country in the list of countries in the world; and $r$ is a region in Africa, Asia, Europe, North America, Oceania or South America. Eq 2b shows computation of $G$.

$$X_{v,\,r}^i = \sum_c X_{v,\,r,\,c}^i \tag{2a}$$

$$G_{s,\,r}^i = \frac{X_{v,\,r}^i}{\sum_r X_{v,\,r}^i} \tag{2b}$$

This approach is used to obtain the distribution of the water availability and demand for various sectors for different regions. Here, the effect of regional climatic conditions is not explicitly considered; rather, the regional distribution of renewable water sources is used where these effects are implicitly accommodated. The agricultural water demand is a function of the area under agriculture in that country. Similarly, the municipal water demand of a country can be computed as a function of its population. In both of these cases, local differences in the water use efficiency in the agricultural sector and municipal water use intensity in that region are not considered. The industrial and energy sector water demands are computed as a function of that country's GDP (measured in international dollar 2011 and adjusted for inflation). Eq 3 is used to compute the regional sectoral water demand $Rs$ for region $r$ and sector $s$.

$$Rs_{s,\,r}^i = Ts_s^i \times G_{s,\,r}^i \tag{3}$$

Total regional water demand $Tr$ can be computed by aggregating the regional sectoral water demand $Rs$ for all sectors for a particular region as shown in Eq 4.

$$Tr_r^i = \sum_s Rs_{s,\,r}^i \tag{4}$$

To compute the regional water stress, regional water availability $A$ is required. It can be computed using Eq 5.

$$A_r^i = M_{WR}^i \times Ga_r \tag{5}$$

where, $M_{WR}^i$ represents global water availability at timestep $i$ and $Ga_r$ represents geographical distribution factor for availability for region $r$ at timestep $i$.

Now, using outcomes of Eqs 4 and 5, regional water stress $Ws$ can be computed using Eq 6.

$$Ws_r^i = \frac{Tr_r^i}{A_r^i} \times 100 \tag{6}$$

As the quantity of available water is finite, it needs to be distributed among various water consumption regions and sectors. The regional and sectoral mass balance are shown in Eqs 7a and 7b, respectively.

$$M_{WR}^i \geq \sum_s Tr_r^i \tag{7a}$$

$$M_{WR}^i \geq \sum_s Ts_s^i \tag{7b}$$

Table 1 summarizes the variables used in this work.

A schematic representation of the working of the water model of the GGSM is shown in Fig 2. On a global level, state variables are used with the sectoral intensities to obtain the total global sectoral demands. Geographical distributions for each of the sectors is used to disaggregate the total sectoral water demand into regional sectoral water demand. At this stage water demands for all the sectors and regions are available. Now, for each region all the sectoral demands are aggregated to get total regional water demand. Then, using the water availability for that region, regional water stress can be computed.

**Table 1. Variable description.**

| GGSM variable/compartment | Symbol |
|---|---|
| Carnivores | C1, C2 |
| Energy production | EP |
| Fuel source | FS |
| Herbivores | |
| Livestock | H1 |
| Feral | H2, H3 |
| Inaccessible resource pool | IRP |
| Industrial sector | IS |
| Inaccessible water reservoir | IWR |
| Primary producers | |
| Agriculture | P1 |
| Grasslands | P2 |
| Forests | P3 |
| Resource pool | RP |
| Water reservoir | WR |
| **Water modelling variables** | |
| Availability of exploitable water (Regional) | A |
| Geographical distribution of water demand | G |
| Geographical distribution of water availability | Ga |
| Availability of exploitable water (Global) | $M_{WR}$ |
| Regional Sectoral water demand | Rs |
| Total Regional water demand | Tr |
| Total Sectoral water demand | Ts |
| Water Stress | Ws |
| Representative variable | X |
| State variable | Y |
| Sectoral Intensity | $\Psi$ |
| **Subscripts** | |
| Country | c |
| Africa, Asia | |
| Region in | r |
| Europe, North America, | |
| Oceania, South America | |
| Agricultural, Livestock | |
| Sector in | s |
| Industrial, Energy | |
| Municipal | |
| Agricultural area, | |
| Meat production, | |
| Representative variable in | v |
| GDP, | |
| Population | |
| **Superscripts** | |
| Timestep | i |

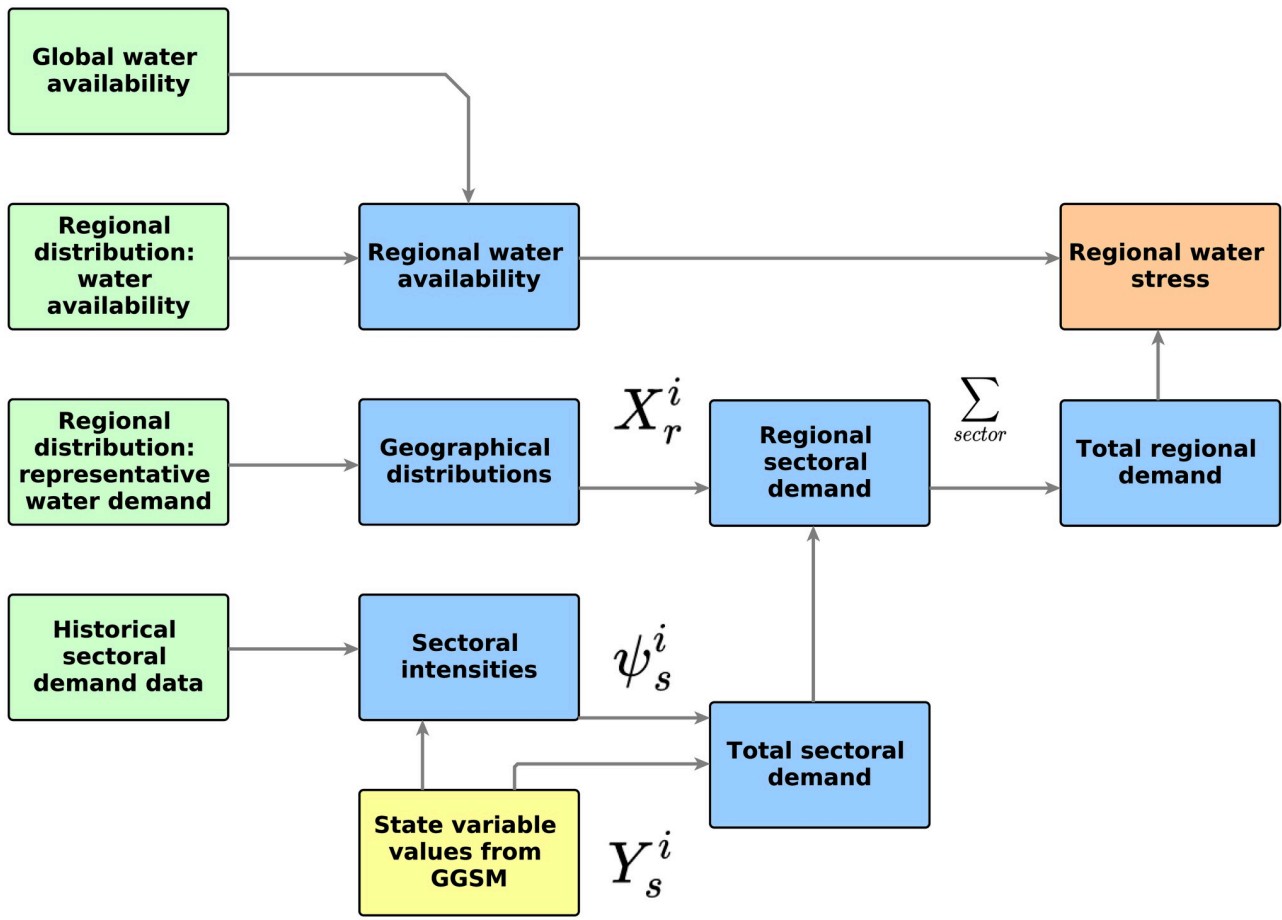

**Fig 2. Schematic representation of water system modelling:** Green colored boxes represent historical/literature based data, blue boxes represent interim variables, yellow box represents model outcomes based data and orange box shows the final outcome.

The working of the water model can be understood through the following example: For the industrial sector, industrial production would be used to obtain the global industrial water demand. Using geographical distribution, regional IS demand is obtained. For a region, for example, Africa, total regional demand is computed from regional demands for agriculture, energy, municipal and industrial demands. This total regional demand is used to compute the water stress for Africa. In the context of supplying these demands, the water reservoirs which represent the stock of exploitable water are utilized. Here, specific modes of demand-supply are not considered, and the competition between sectors is ignored. A simple water balance based on material flow analysis is employed. The water stress is computed for all regions and sectors. If the water stress is > 100%, it represents a situation where non-renewable water would be used.

## Model parameterisation and validation

In this work, the variable trends and parameter values are obtained by fitting historical water demand and availability data from 1950 to 2013. The projections are obtained till the year 2120 with a time step of one week. Based on data from AQUASTAT [19], the total global exploitable water ($M_{WR}^i$) and the total renewable water resources are about 135 and 1060 billion

cu m per week, respectively. These quantities are assumed to be constant over the simulation horizon for the current work. Inaccessible water is the difference between total renewable water and total exploitable water. Hence, inaccessible water resources ($M_{IWR}^i$) are 925 billion cu m per week.

**Global sectoral demand parameterization.** The sectoral intensity trends are used to compute the water demand for a sector using GGSM variable values. In other words, they represent the transformation functions to obtain the water demand for a particularSince the new figure was added at location 2, the names of other figure files had to be updated. The complete set of images is attached herewith. sector from corresponding state variables. The historical sector-wise water demand data is obtained from the AQUASTAT database [19]. This country-level data is then aggregated continent-wise and mapped against state variable data ($Y^i$) for the same period to obtain the sectoral intensity plots ($\Psi$). Historical data is explicitly available for agricultural (*P1*), and municipal (*HH*) sectors; however, separate historical data for the industrial (*IS*) and energy (*EP*) sectors are not available as the water withdrawal data for the industrial sector also includes the energy sector. Hoekstra [24] analyzed the global water consumption by various sectors. From 1996 to 2005, they computed the combined industry and energy sector water demand to be 400 billion cu m per year. Mekonnen *et al.* [25] have computed annual global energy water for the period 2000–2005 to be around 250 billion cu m. Using this information, 62.5% of the aggregate industrial water demand can be allocated to the energy sector and 37.5% to the industrial sector. It is assumed that this allocation remains constant over the simulation horizon. The sectoral intensity for the livestock (*H1*) sector is linearly proportional to its mass.

The country-level data is aggregated for each continent, and piece-wise linear fits [26] are obtained. The segments are selected to capture the effect of an increase in the water use efficiency as the demand volume increases. These are depicted in Fig 3. These fits are sectoral

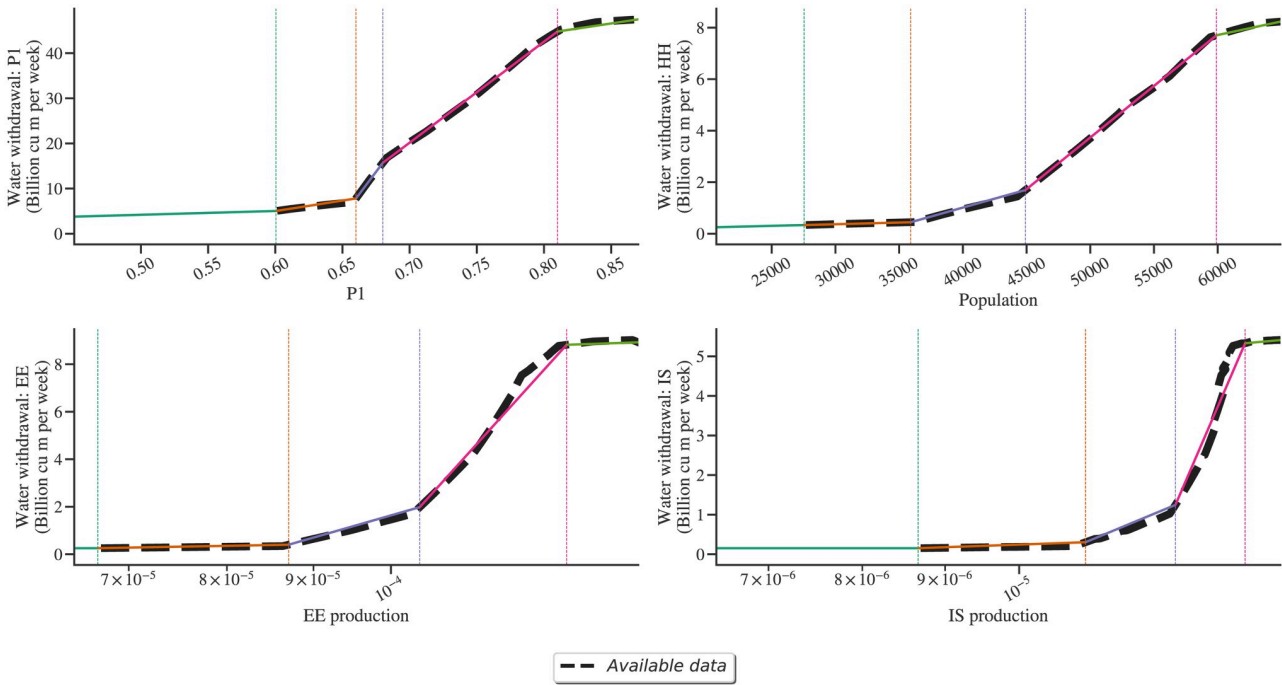

**Fig 3. Sectoral intensity trends: The water withdrawal data (y-axis) is plotted against corresponding sector from the GGSM (x-axis).** Light green thick dashed lines show these trends. Exact equations are shown in Section S1 in S1 File.

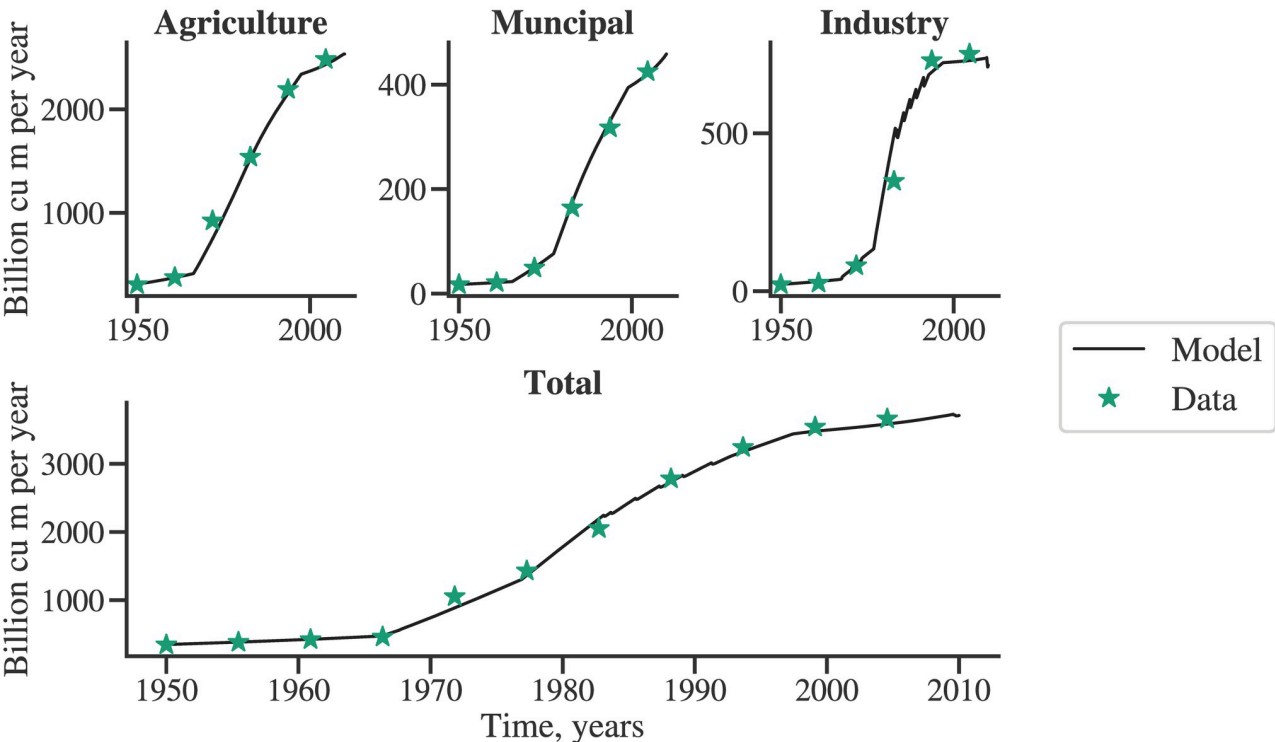

**Fig 4. Validation of the sectoral water demand with respect to historical data.**

intensity trends. The use of these trends can be elucidated with the example of the agricultural sector. When P1 (state variable value) is 0.75, the global agricultural water demand is about 32 billion cu m. Other sectors can also be read in the same fashion.

Fig 4 shows the validation of the global sectoral water demand computation model. The markers indicate the historical global water demand for agriculture, industry, and municipal sectors. The historical data of industrial sector water demand also includes the energy sector. Accordingly, the model outcomes' global water demand of the energy and industrial sectors are aggregated before the comparison. The global water demand computed by the model for each sector, agriculture, municipal, and industrial (also including energy), conform well with the historical data.

**Regional sectoral demand parameterization.**  Based on data from Food and Agriculture Organization, United Nations [27], the regional distribution of the availability ($Ga_r$) of water is as follows: Africa: 9%, Asia: 28.4%, Europe: 15.2%, North America: 17%, Oceania: 2.1%, and South America: 28.3%. The demand distributions for the agriculture and livestock sector are obtained using representative variables as agricultural area per capita [28] and meat production [29], respectively. For the industry and energy sector, real GDP per capita [30] is used as a representative variable. Population data is used to obtain the distribution of the municipal water demand [31]. Several nation-states came into existence due to significant global political events in the early 1990s. As information related to such countries before 1992 is not available consistently, country-level data from 1992 to 2013 is used. First, the per capita agricultural area and per capita GDP are converted to total values using the population data. This historical country-level data is now extrapolated to the requisite time horizon. It is ensured that the agricultural area does not exceed its total land area. Then, countries corresponding to different

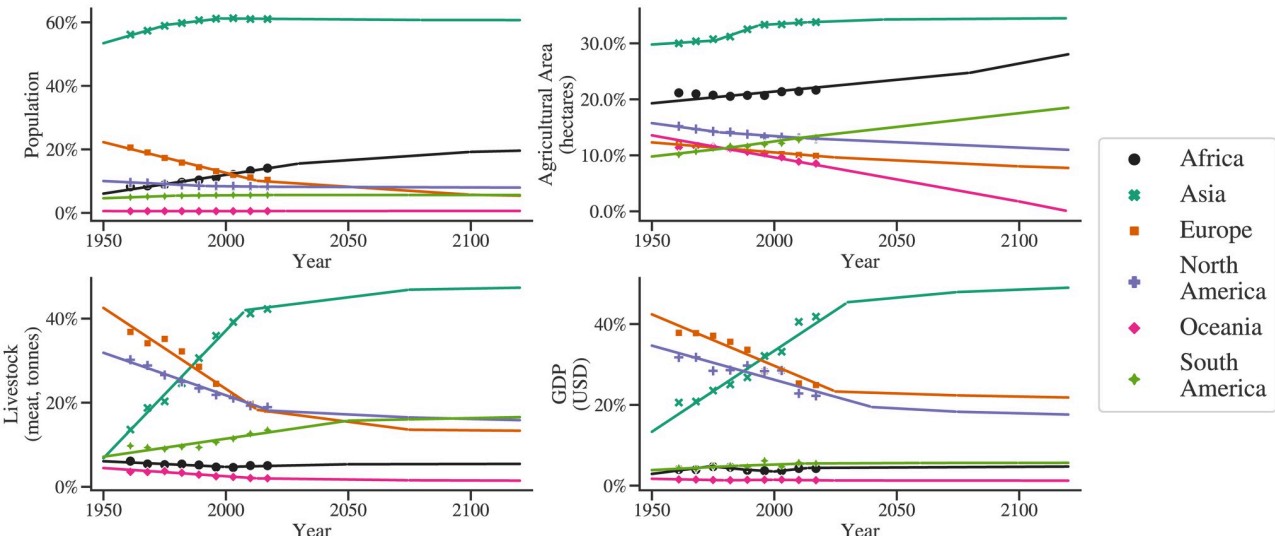

**Fig 5. Geographical distribution of the water demand: The trends of regional values of the representative variables are extrapolated to the year 2120.** The contribution of each region towards the global value of each of the representative variables is depicted here. The historical data is shown by the markers, and the solid lines show the fits to historical data and the predicted data both. Detailed procedure of regional modelling in GGSM is provided in Section S2 in S1 File.

groups (here, continents) are aggregated. Due to the discontinuous nature of the data, the approach of piece-wise linear fit [26] is implemented to obtain the linear regression fits. Fig 5 shows the regional distribution projections for different sectors. The historical data is also shown in this figure, and hence, this comparison serves as a validation for the regional water demand distribution model.

Table 2 shows the summary of the model parameters and Table 3 summarizes the representative variables for various sectors.

## Scenario planning

Similar to the works of Nisal *et al.* [15], simulations are carried out for the period of 1950–2120. This simulation horizon can help identify the long-term consequences of the business-as-usual and scenarios with perturbations. For the business-as-usual or the base case scenario,

**Table 2. Summary of model parameterization.**

| Parameter details | Values | Source |
|---|---|---|
| Historical water demand and availability | $M^i_{WR}$: 135 billion cu m/week<br>$M^i_{IWR}$: 925 billion cu m/week<br>Agricultural, industrial and municipal water withdrawal | AQUASTAT [19] |
| State variable values | Mass of P1, H1; Production of IS, EE; Population | Nisal *et al.* [15] |
| IS/EP split of industrial water demand | 37.5%/62.5% | Hoestra [24]<br>Mekonnen *et al.* [25] |
| Regional availability distribution | Africa: 9%,<br>Asia: 28.4%,<br>Europe: 15.2%,<br>North America: 17%,<br>Oceania: 2.1%, and<br>South America: 28.3% | FAO, United Nations [27] |

**Table 3. Representative variables for water demand from different sectors.**

| Sector | Representative variable | Source |
|---|---|---|
| Agriculture | Agricultural area | [28] |
| Livestock | Meat production | [29] |
| Industry and Energy | GDP | [30] |
| Municipal | Population | [31] |

a medium population growth rate is considered [32]. Unless explicitly mentioned otherwise, population growth projections of the base case are used. The consumption coefficients of the base case are used for all the studies such that there is no significant increase in the demand for the goods by human society.

1. Baseline demand-availability dynamics: The first part of this study focuses on the global scale. The aim is to analyze the temporal dynamics of the water stress and contributions by various sectors and regions. Then, each sector's streams of water withdrawal are analyzed with respect to the contributions of the regions and vice versa. These studies help understand the global water scenario and identify the regions and sectors critical from a water crisis point of view.

2. Population explosion: The first study uses a medium growth rate of the global population. Increasing population growth is a critical challenge to global sustainability. The UN population projections provide the different population growth rate variants, namely, lower, medium, and upper. Interested reader may refer to the work by Nisal *et al.* [15]. The high population growth rate projections have been used to model this scenario. A comparison between the base case and the population explosion case can highlight the severity of the water crisis through the prism of the population.

3. Mitigatory measures: Agenda 2030, agreed upon by the international community in 2015, recognized the criticality of water resources. Sustainable Development Goal (SDG) 6 enshrines the stakeholders with several responsibilities related to the global water sector. These goals include 6.2 and 6.4, focusing on improving water use efficiency. Any progress in these fields would help reduce the water demand and hence the water stress. This study aims to model improvement in water use efficiency and quantify its benefits.

Nisal *et al.* [15] analyzed the scenarios of the population explosion and consumption increase. In this work, three studies are undertaken: first, using the revised model to analyze the demand-availability dynamics across different regions and sectors; second, investigate the implications of the population explosion scenario; and third, assess the efficacy of the mitigatory measures.

## Results and discussion

This section reports the results for the three scenarios mentioned in the previous section.

### Baseline demand-availability dynamics

Fig 6 presents the global water demand-availability dynamics. The top row shows absolute water demand values, and the bottom row shows water stress, that is, water demand relative to availability. The projected water demands of different sectors and continents are presented in the left and right column plots. The contribution of the cumulative livestock and agricultural sector, cumulative industry and energy sector, and municipal sector towards the total water

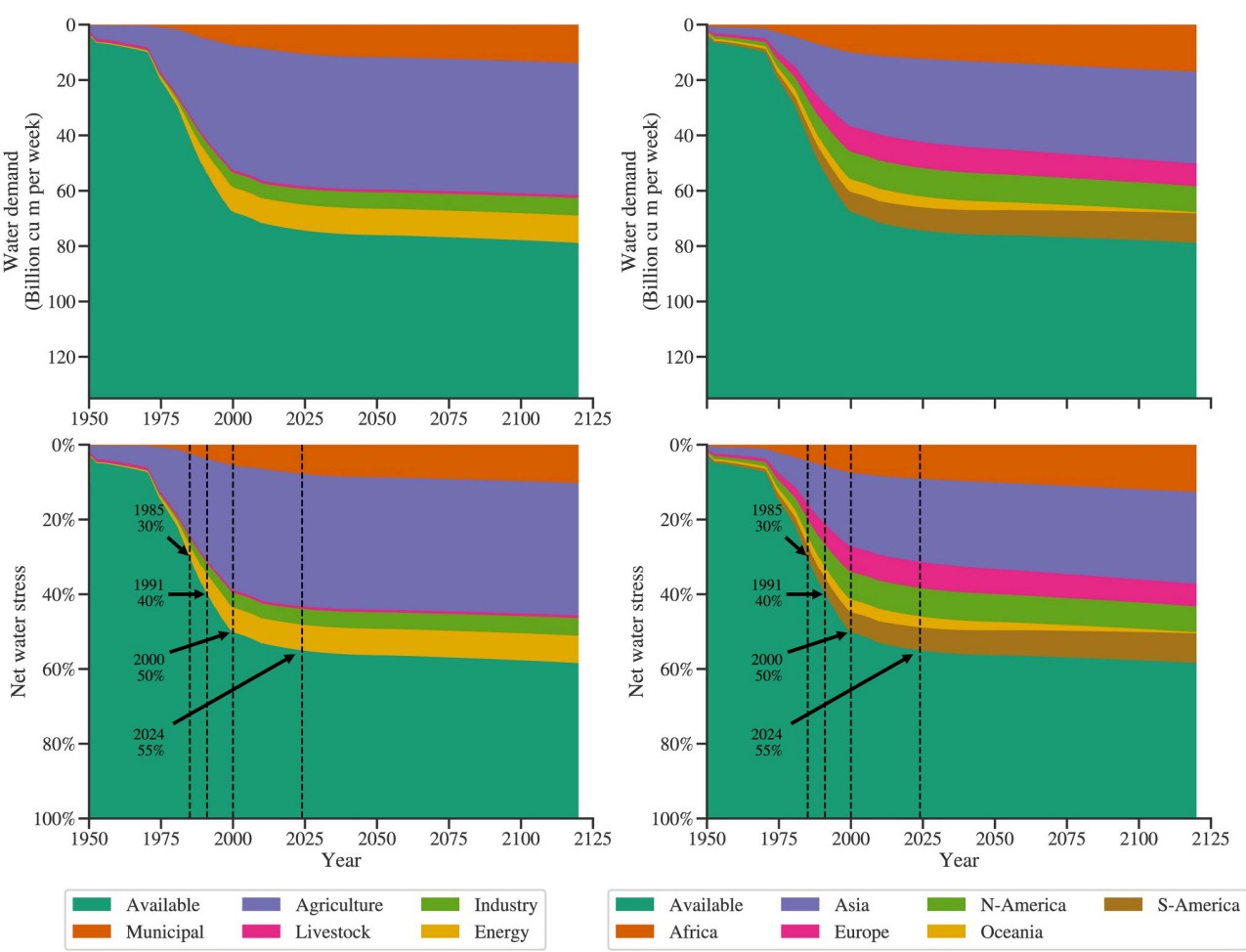

**Fig 6. Global demand availability analysis: The top row plots show water demand projections' absolute values, whereas the bottom row plots show the relative demand expressed in percentage.** The left column depicts the contribution of different sectors to total global water demand, whereas the right column shows a similar distribution for various regions.

withdrawal for the year 2014 is about 70%, 20%, and 10%, respectively. This agrees with the information provided by the AQUASTAT [33]. Simulation results show that global average water stress could reach 55% in 2024; however, till the year 2120, it does not exceed 60%. Hence, adequate water is available on a global level to address the rising demand. This can be attributed to the gradually increasing water demand by the agricultural sector and the reduction in the population growth rate. As the agricultural sector grows, owing to the efficiency increase captured by the sectoral intensity trends, the marginal requirement of water reduces. Consequently, the agricultural water demand increase is relatively lower. The population growth rate, modeled taking cognizance of the medium variant of the UN population growth projections, reduces soon after 2020. As a consequence, the growth rate of the municipal water demand also reduces relatively. The global demand for the livestock sector is observed to be insignificant as compared to other sectors. This is because, in this model, water demands are categorized as per the compartments. Much of the water footprint of livestock comes from agriculture that provides feed to livestock. Here, water demand is considered in demand by the P1 (agriculture) sector, and the direct water demand by H1 (livestock) is low.

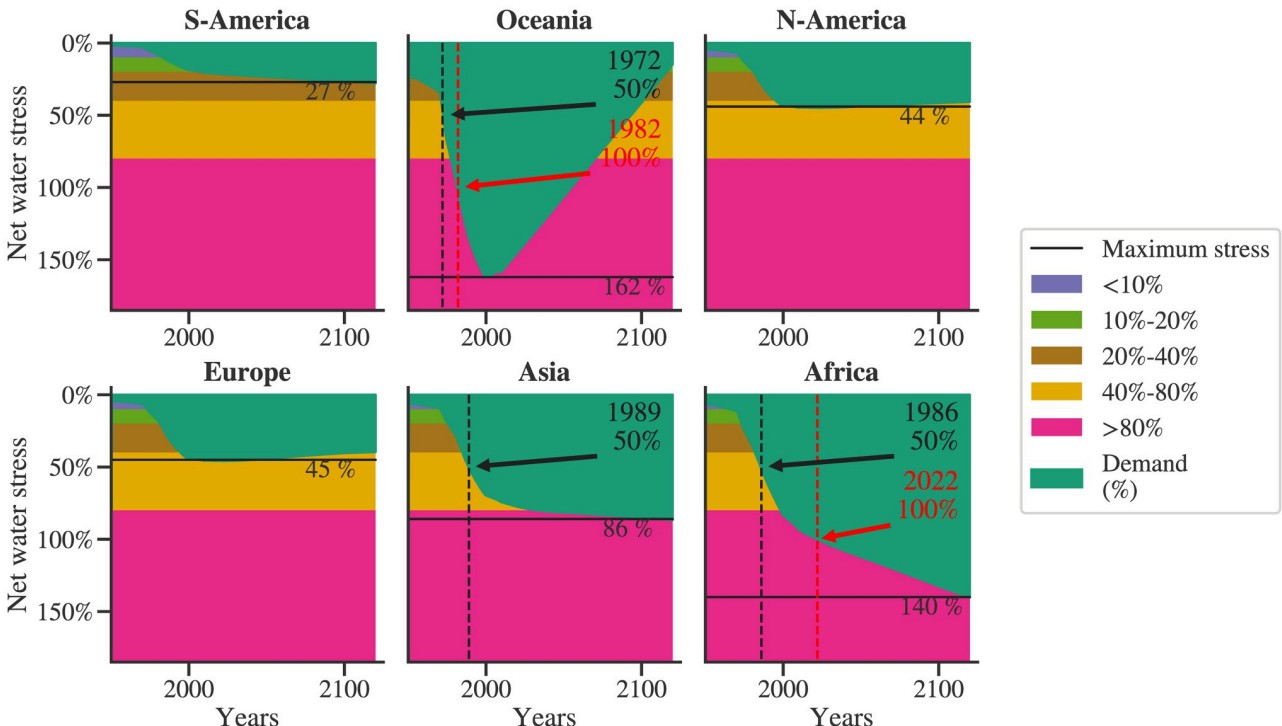

**Fig 7. Regional water stress dynamics: Different levels of water stress as identified by the World Resources Institute are shown.** Specific timesteps are identified if and when the demand for a region crosses a threshold of 50% and 100% of its availability. The maximum value of the water stress is shown.

These results are compared with the numbers reported in the literature. According to United Nations [34], the global water stress is 17% for the year 2017, which is significantly lower than the results obtained from this investigation (50–55%). The United Nations and the World Resources Institute [35, 36] used the total renewable water resources to represent the availability of the water. It includes all the renewable water resources taking into consideration the water required for environmental processes. This approach does not take into consideration the economic or technological limitations on utilizing the water available. In other words, the water stress computed by this approach can be termed gross water stress. On the other hand, this investigation uses exploitable water to compute the water stress, and hence the water stress value is higher. We believe that the values calculated here are more representative of the actual water stress.

The right column plots show the contribution of different regions. Asia and Africa are observed to be the leaders in water demand. The contribution of water demand to the total by economies in a developed state today is North America, and Europe is expected to decline because of their reducing GDP and population (see Fig 5). More insights in this context can be obtained by analyzing the water stress for each region.

Fig 7 shows the water stress dynamics for different regions. The time at which water stress crosses certain thresholds is marked in the figure. The maximum water stress for each region is also identified. South America, Asia, and Africa show increasing water stress. The water stress for Europe and North America is observed to reduce marginally. Past water crisis events such as the one in Cape Town led to the definition of 'Day Zero' [37, 38]. 'Day Zero' is defined as the day in a particular year when the city's water availability falls below 13.5%. On similar

lines, here we define 'Year Zero' where the total projected water demand becomes equal to water availability. A key takeaway message from this analysis is that—Asia and Africa are already facing substantial water stress, and extreme water crises are expected in these regions in the future to come. The net water stress in Africa will reach 100% in 2022; that is, the 'Year Zero' for Africa might reach very soon. The net water stress in the Oceania region increases drastically and is seen to reach about 162%. We acknowledge that this exaggeration is due to the water modeling limitations. Because of the relatively small proportion of the Oceania region, the impact of small inaccuracies gets magnified significantly. The particular simplification leading to such behavior is the aggregate modeling of the water requirement by the agricultural sector. Here, the water demand by agriculture is modeled as a homogeneous quantity that is proportional to the area under agriculture. In reality, this water requirement is heterogeneous and depends on the type of crop being produced. As the Australian continent hosts about 10–12% of the global agricultural area, its water demand is very high. Wheat and barley are the crops that dominate Australian agriculture. As these are low water-consuming crops, the real water stress is expected to be lower than the numbers projected here. However, there are legitimate concerns regarding the water scarcity in Australia [39]. A more thorough understanding of the causality behind such projections can be obtained by analyzing the contribution of different sectors to the total regional demand.

Fig 8 shows the contribution of each sector to the water demand by each of the regions. Two distinct regimes can be seen here, first from 1950-the 1990s, and second from 1990 onwards. In the first regime, the agricultural sector is a dominant water consumer across all the regions. However, its importance is expected to reduce relative to other sectors for Europe and Asia, with agricultural demand contribution falling below 50% of the total demand in

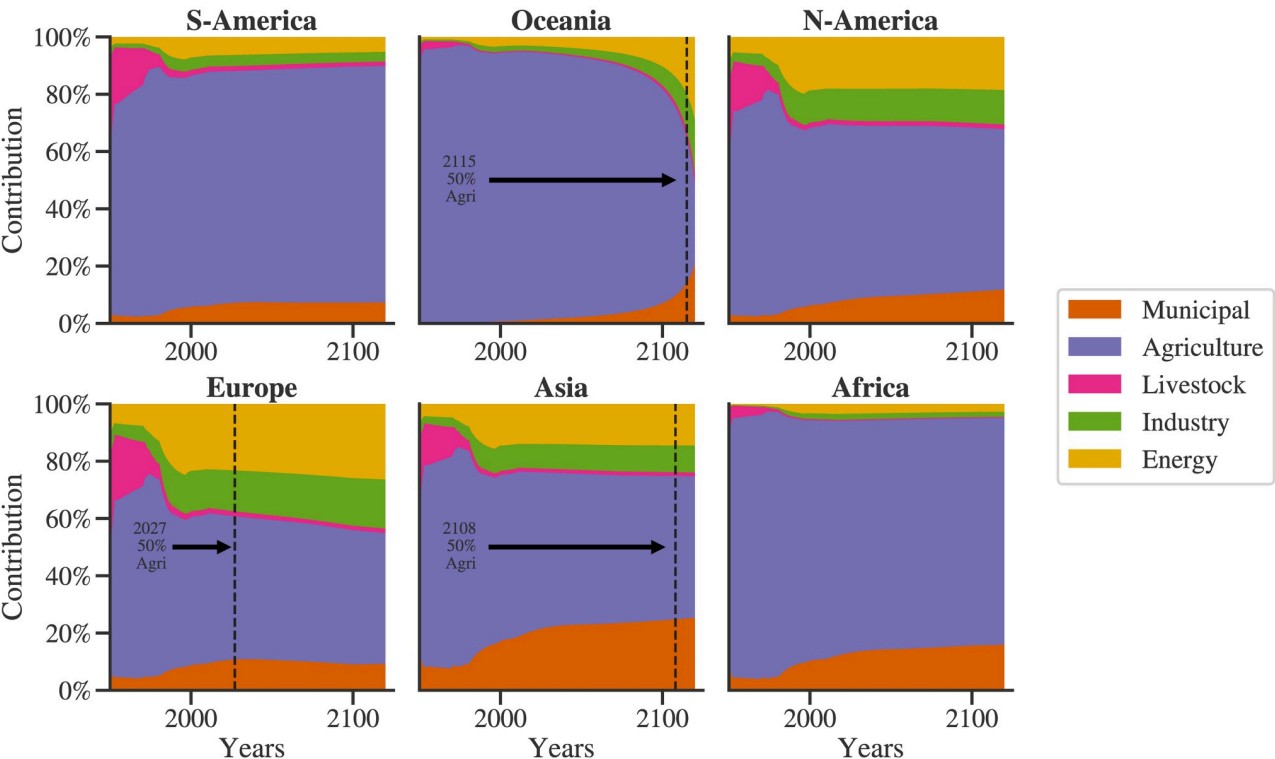

**Fig 8. Regional- sectoral water demand analysis: Specific timesteps are identified, if and when the contribution of agricultural demand for a region crosses 50%.**

2027 and 2108, respectively. As discussed earlier, this model captures the increase in water use efficiency with increasing agricultural production. Hence, other sectors start dominating the regional demand. With the rapid increase in the population in the first regime, the overall global water demand increased. Consequently, in the initial period, the contribution of livestock is significant and reduces as total water demand increases. The water demand by the industry, energy, and municipal sectors as well is affected by the population growth rate. This effect is reflected in their contribution trends which show an increase in the initial period and then stabilize. For Europe, the municipal demand has a declining trend owing to the reducing population. On the contrary, the contribution of municipal demand towards total regional demand for Asia is increasing. In light of the overall demand for Asia and Europe, as seen in Fig 7, municipal demand is identified as the driving component following agriculture.

Oceania consists of island countries such as Fiji, French Polynesia, Australia, New Zealand, and so on. Figs 5, 8 and 9 show that the trends of the agricultural sector of Oceania are declining. As a result, as seen in Fig 8, the agricultural sector's contribution to its total water demand reduces and eventually falls below 50% at the year 2115. Fig 9 depicts contributions of various regions to each of the sectors. This figure also brings out another observation, water demand for agriculture is roughly evenly distributed among the regions; however, for others, Asia is the most dominant region from the water demand point of view.

Fig 9 shows the contribution of each region to the sectoral demand. Due to its high population, municipal water consumption is dominated by Asia, with a significant margin. However, though Asia uses a large portion of its land for agricultural purposes, other regions also contribute towards agricultural water demand in a significant proportion. Similar observations can be made for the energy, industry, and livestock sectors. This particular plot can be useful for agencies specializing and leading in respective sectors; for example, International Energy Agency (IEA) for energy; United Nations Industrial Development Organization (UNIDO) for industry; and Food and Agricultural Organization (FAO) for agriculture and livestock. Such agencies can use these projections to identify the priorities for policymaking.

Some perturbations can be seen in the contributions of various sectors towards a regional water demand in a period from the 1980s to 2000s (Fig 8). In contrast, no such perturbations can be observed in Fig 9. This indicates the regional development in different sectors and related dynamics captured by the model. The contribution of various regions towards the total sectoral water demand changes more slowly than the contribution of various sectors towards the total regional water demand. This indicates that changes in the demand dynamics within a sector have more inertia than demand dynamics of a region. Thus, it is relatively easier to influence the water withdrawal patterns in a region than to modify the water withdrawal of a sector on a global level. In other words, in altering the demand patterns of a region, the policymaking entities are expected to face fewer challenges than altering the demand dynamics of a sector globally. Though this is an intuitive outcome, we have supported it with historical data and rigorous modeling.

### Population explosion scenario

The simulations carried out so far used the medium growth variant of the UN population projections [32]. However, population explosion is considered one of the significant global challenges. The upper growth variant of the UN population projections is used to model the population explosion scenario. A comparison of the population explosion and the base case is shown in Fig 10. The population explosion water stress profile follows the same trend as the base case and then clearly diverges from it from 2030–2050. Following this diversion, the maximum stress in the population explosion scenario increases for the populous regions compared

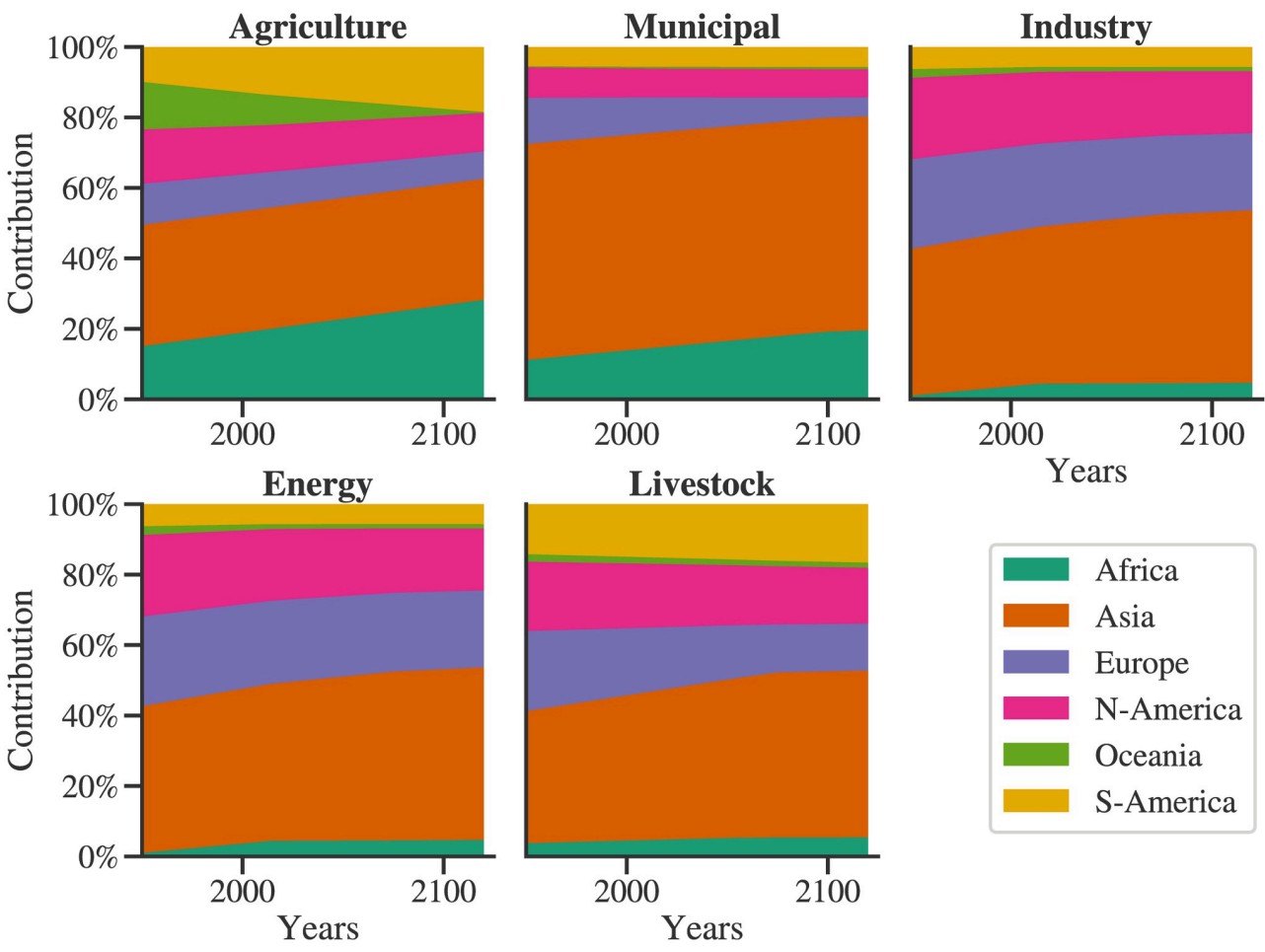

**Fig 9. Contribution of regions to the total demand by the sector.**

with the base case. For Asia, water stress increases from 86% to 97%, for Africa from 140% to 147%, and for South America from 27% to 28%.

## Mitigatory measures

The global, sectoral, and regional dynamics of the water stress for the base case and the population explosion have been analyzed so far. These studies point towards impending regional water crises. Sustainable development goal (SDG) 6, which focuses on water, has two specific objectives improving water management and water use efficiency. Progress in these fields is expected to reduce water use compared to the case where there is no progress. Fig 11 compares these two cases, namely, the base case and the one with a 30% reduction in the water consumption of the agricultural sector. The water use efficiency of the agricultural sector is assumed to increase linearly from 2015 and reach a maximum in the year 2030. The maximum water use efficiency is assumed to reduce the agricultural water requirement by 30% compared to the base case. These modifications lead to significant benefits. The Year Zero for Africa is observed to be shifted from 2022 to 2097. For Asia, the maximum water stress reduces from 86% to 75%. However, given the model assumptions and expected uncertainties, 75% water stress is

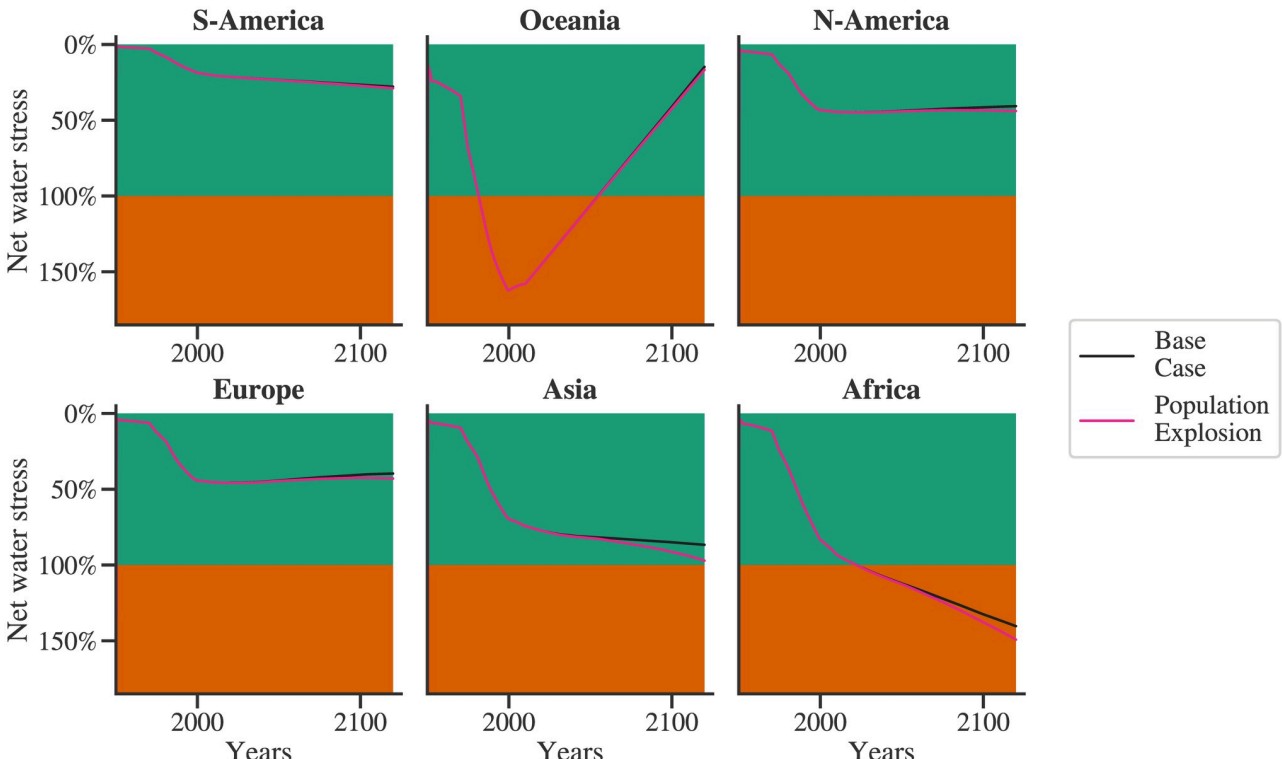

**Fig 10. Population explosion: A comparison of water stress for the base case and population explosion case for the six regions is shown here.**

still a concern. Therefore, the simulation results indicate that improvements only in the agricultural sector will not be enough to manage water stress in Asia. Either greater improvements in agriculture are needed, and/or similar measures need to be undertaken to reduce municipal water consumption, which is the next major contributor. Improvements in agricultural water use efficiency also benefit South America, with a reduction in stress from 27% to 21%. For other regions, North America, Oceania, and Europe, though the maximum stress is not significantly influenced, efficiency improvement results in lower water withdrawal and economic benefits.

## Conclusion

Understanding of systemic dependencies of the components of the FEW nexus is crucial. As a step towards developing this understanding, the GGSM is used to model the global water system and explore the demand-availability dynamics, that is, the water stress, on a global and regional scale. The proposed net water stress is more realistic than the current water stress indicator. Based on the results, it can be concluded that though global water availability is adequate to satisfy the water demand on an aggregate level, a water crisis might occur in the future on a regional level. Asia is identified as the potential flashpoint of the impending water crisis. A higher population growth rate would worsen the situation for Asia. The agricultural sector governs the water demand across the continents. The water use efficiency improvement for the agricultural sector, which would reduce the water demand by 30%, could significantly delay this crisis. Although these results agree with previous predictions and expectations, the model results provide quantitative information. Moreover, the model also allows testing

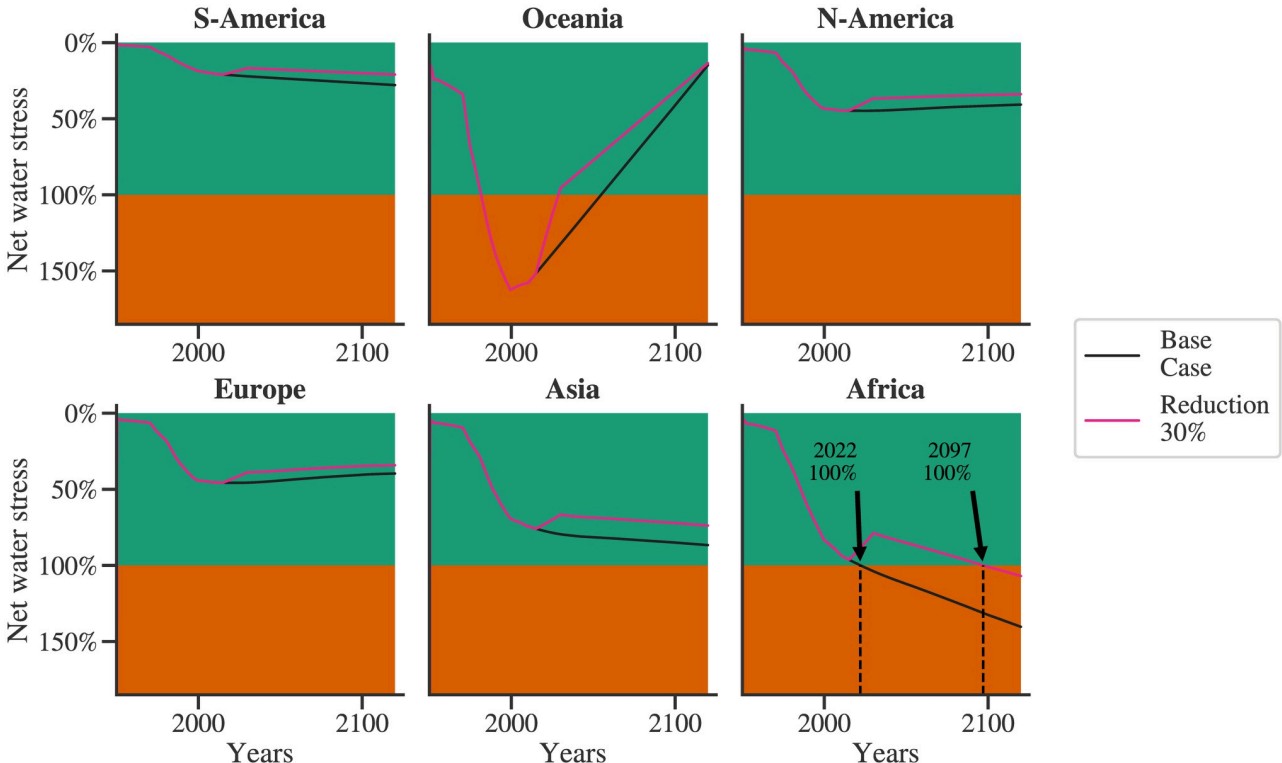

**Fig 11. Mitigatory measures: A comparison of base case and 30% reduction in water demand of the agricultural water demand owing to the achievement of SDG 6.2 and SDG 6.4.**

various mitigation strategies and quantifying their benefits. From that view, the model can be a key contributor to sustainable development policymaking. Many such alternatives will be explored as a follow to this work.

This work was subject to several assumptions and simplifications, which also form a basis for potential avenues for future research. These are as follows:

1. This work focuses only on the demand side. It does not delve into detailed modeling of the recycling of water which is a critical factor in the context of high water stress region. Moreover, often the water use efficiency improvement is at the expense of energy resources. Hence, the dynamics between water recycling in water-stressed regions and energy requirements can shed light on a trade-off between efficiency improvement and water recycling.

2. Secondly, the economics of water is not considered. This is a crucial limitation. Incorporation of water pricing will permit modeling the impact of stress on water demand, thereby providing greater insights.

3. The influence of climate change is another factor that is beyond the scope of the current work. Regional as well as global precipitation patterns can change because of climate change. Climate change may also influence the crop patterns, thus, altering the water intensity of the agricultural sector. Hence, one of the potential future research avenues could be the modeling the climate change influences on water demand and availability.

4. Lastly, as a simplification, the regional aggregation of countries is carried out based on the continent in which that they are part of. However, a better approach would be grouping the

countries based on the river basins they are sharing. Another simplification is aggregate modeling of the agricultural water demand. Incorporation of different crops and their water demand could make the model outcomes more realistic.

## Supporting information

**S1 File. Section S1**: Sectoral water intensity equations are presented in this section. **Section S2**: This section describes the method adopted for modelling regions in this work. (PDF)

## Acknowledgments

This is a collaborative project between the USA, India, and Hungary. The authors would like to thank R. Boumans for sharing data from the Global Unified Model of the BiOsphere (GUMBO) and his invaluable inputs from the same.

## Author Contributions

**Funding acquisition:** Yogendra Shastri.

**Investigation:** Neeraj Hanumante, Yogendra Shastri, Apoorva Nisal, Urmila Diwekar, Heriberto Cabezas.

**Methodology:** Yogendra Shastri, Urmila Diwekar.

**Project administration:** Urmila Diwekar.

**Software:** Neeraj Hanumante.

**Supervision:** Yogendra Shastri.

**Validation:** Neeraj Hanumante.

**Writing – original draft:** Neeraj Hanumante.

**Writing – review & editing:** Yogendra Shastri, Urmila Diwekar.

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
