## [Editor Report · Decision Letter 0]

13 Sep 2021

PONE-D-21-28130Integrated model for Food-Energy-Water (FEW) nexus to study global sustainability: The water compartments and water stress analysisPLOS ONE

Dear Dr. Urmila,

Thank you for submitting your manuscript to PLOS ONE. I have checked your manuscript and I am confussed about it. My first concern is that the work you present here is an extension to another paper already submitted to our journal. You must clarify what the need for a separate paper for an extension to the same model is. Indeed, this is something that it had been better to explain to the journal office during the submission to try to have the same editor for both works. Also, both manuscripts seem to share part of the authors.

Therefore, please, justify the reason to split both works. In the meantime, I will contact the office to discuss this issue.

Secondly, from the current title of the manuscript, it seems that you present a new model. But you do not include any code for it (at least I have not found it after a quick read). In the list of Supporting Information, you mention the equations of the model. Probably you have not computed your results by hand, but a computer implementation of them. You have to include it with the text. The same applies to the outputs of the model, the simulations that you mention. All this information is necessary to ensure the reproducibility of your work by reviewers and potential readers. Also, what would be the point of presenting a new model and limiting its use by others?

We look forward to receiving your revised manuscript.

Kind regards,

Juan A. Añel

Academic Editor

PLOS ONE

Journal Requirements:

2. Please note that PLOS ONE has specific guidelines on software sharing (http://journals.plos.org/plosone/s/materials-and-software-sharing#loc-sharing-software) for manuscripts whose main purpose is the description of a new software or software package. In this case, new software must conform to the Open Source Definition (https://opensource.org/docs/osd) and be deposited in an open software archive. Please see http://journals.plos.org/plosone/s/materials-and-software-sharing#loc-depositing-software for more information on depositing your software.

"This is a collaborative project between the USA, India, and Hungary. Collaborators from India acknowledge support from the Ministry of Human Resource Development, Government of India, through the SPARC (project code: P1238). The research contribution by H. Cabezas was carried out in the GINOP-2.3.2-15-2016-00010 framework “Development of enhanced engineering methods with the aim at utilization of subterranean energy resources” project at the Research Institute of Applied Earth Sciences of the University of Miskolc, the Sz´echenyi 2020 Plan, partially funded by the

European Union, co-financed by the European Structural and Investment Funds. The authors would like to thank R. Boumans for sharing data from the Global Unified Model of the BiOsphere (GUMBO) and his invaluable inputs from the same. "

"NH, YS, UD acknowledge the support from the Ministry of Human Resource Development,  Government of India, through the SPARC (project code: P1238). 

The research  contribution by H. Cabezas was carried out in the GINOP-2.3.2-15-2016-00010  framework Development of enhanced engineering methods with the aim at utilization  of subterranean energy resources" project at the Research Institute of Applied Earth  Sciences of the University of Miskolc, partially funded by the European Union, co-financed by the European Structural and Investment Funds.
---

## [Author Response · Author response to Decision Letter 0]

21 Sep 2021

Dear Editor,

Thank you for your quick review of our papers. I am providing the rebuttal with this letter. We are also submitting the revised manuscript where we have taken out the funding section from the acknowledgment and added the link for accessing the code. The funding format you have provided looks good; however, we have slightly modified it as follows. 

Thanks, and looking forward to your response. 

Regards

Urmila Diwekar

FUNDING PARAGRAPH:

NH, YS, UD acknowledge the support from the Ministry of Human Resource Development, Government of India, through the SPARC (project code: P1238). 

The research contribution by H. Cabezas was carried out in the GINOP-2.3.2-15-2016-00010 framework Development of enhanced engineering methods with the aim at utilization of subterranean energy resources" project at the Research Institute of Applied Earth Sciences of the University of Miskolc, partially funded by the European Union, co-financed by the European Structural and Investment Funds.

The funders had no role in study design, data collection, and analysis, decision to publish, or preparation of the manuscript."

REBUTTAL:

Editor’s comment: My first concern is that the work you present here is an extension to another paper already submitted to our journal. You must clarify what the need for a separate paper for an extension to the same model is. Indeed, this is something that it had been better to explain to the journal office during the submission to try to have the same Editor for both works. Also, both manuscripts seem to share part of the authors.

Therefore, please, justify the reason to split both works. In the meantime, I will contact the office to discuss this issue.

Response: The focus of the first paper was to describe the global sustainability model in terms of the food web and the microeconomic model. The focus of the first paper was to identify model parameters so as to align model output with historical global values. The model was then used to perform simulations for the next 100 years and study global sustainability in terms of the population of various ecological compartments, economic metrics such as GDP, and track various emissions like GHG and NOx. 

The focus of the second paper, in contrast, is to study global water sustainability. Although the model is based on the one presented in the first paper, it incorporates several modifications as highlighted here:

• Water compartments, as well as the stocks and flows of water, are incorporated.

• The parameter values for the various equations governing the utilization of water by various sectors such as industry and agriculture are estimated using historical data.

• Since water stress is highly region-specific, the model is further modified to consider different continents. Therefore, the water availability, consumption, and resulting stress are calculated and simulated for each continent separately, thereby providing more granular insight. Note that the model in the first paper did not study continent-wise trends. 

Therefore, the approach we have taken in the second paper is different from the first paper. Further, the details involved are crucial, and combining both papers would have caused more confusion. Therefore, we have split the papers. The papers can be judged separately or together. We would appreciate it if the same reviewers judged them.

Editor’s Comment: Secondly, from the current title of the manuscript, it seems that you present a new model. But you do not include any code for it (at least I have not found it after a quick read). In the list of Supporting Information, you mention the equations of the model. Probably you have not computed your results by hand, but a computer implementation of them. You have to include it with the text. The same applies to the outputs of the model, the simulations that you mention. All this information is necessary to ensure the reproducibility of your work by reviewers and potential readers. Also, what would be the point of presenting a new model and limiting its use by others?

Response: We have a large code for this model which we could not submit. However, we are providing a link to google drive where the model currently sits. This drive has all the data, manual, and code. We have now included this in the current manuscript.

---

## [Decision Letter · Decision Letter 1]

10 Dec 2021

PONE-D-21-28130R1Integrated model for Food-Energy-Water (FEW) nexus to

study global sustainability: The water compartments

and water stress analysisPLOS ONE

Dear Dr. Urmila,

Thank you for submitting your manuscript to PLOS ONE. After careful consideration, we feel that it has merit but does not fully meet PLOS ONE’s publication criteria as it currently stands. Therefore, we invite you to submit a revised version of the manuscript that addresses the points raised during the review process.

We look forward to receiving your revised manuscript.

Kind regards,

Juan A. Añel

Academic Editor

PLOS ONE

Reviewers' comments:

Reviewer's Responses to Questions

**Comments to the Author**

1. If the authors have adequately addressed your comments raised in a previous round of review and you feel that this manuscript is now acceptable for publication, you may indicate that here to bypass the “Comments to the Author” section, enter your conflict of interest statement in the “Confidential to Editor” section, and submit your "Accept" recommendation.

Reviewer #1: (No Response)

Reviewer #2: (No Response)

Reviewer #3: (No Response)

2. Is the manuscript technically sound, and do the data support the conclusions?

Reviewer #1: Yes

Reviewer #2: No

Reviewer #3: Partly

3. Has the statistical analysis been performed appropriately and rigorously? 

Reviewer #1: Yes

Reviewer #2: I Don't Know

Reviewer #3: Yes

4. Have the authors made all data underlying the findings in their manuscript fully available?

Reviewer #1: Yes

Reviewer #2: Yes

Reviewer #3: Yes

5. Is the manuscript presented in an intelligible fashion and written in standard English?

Reviewer #1: Yes

Reviewer #2: Yes

Reviewer #3: Yes

6. Review Comments to the Author

Reviewer #1: This is a well-written paper addressing a topical subject of the WEF nexus. The authors have provided every detail including the modelling code. The authors need to add references to some of the statements in the introduction and check the grammar. But otherwise this is a very good study.

Reviewer #2: This paper uses and expands the global “integrated cross-disciplinary model” known as Global Sustainability Model (GGSM) to investigate water sustainability under different future scenario of water demand. The authors claim that this is done while considering the food-water-energy nexus.

The development of models of the food-water-energy nexus is a hot topic with potentially significant societal impacts. That said, I believe that the authors have not provided enough details to understand what exactly GGSM simulates and how. For example:

- How are the different water resources allocated within each country?

- How are “economic and technological considerations” taken into account to assess how much water can be harvested?

- How is water demand of the different sector satisfied?

- How was the model validated (Figs. 2 and 3 are not clear at all)?

- What are the model limitations and assumptions?

- Are the two-way interactions between food, water and energy taken into account? And if yes, how?

I am sorry for the short review, but without these details, I am unable to provide any additional feedback on the results. At this stage, I thus recommend its rejection.

Reviewer #3: The authors present an interesting approach to model the water stress on the continents. However, there are still some parts that make it very hard to understand and therefore unsuitable for publication in its present form.

The section Modeling global water system would improve a lot from a more detailed presentation of the equations and variables. Some problems currently are

• Equation (1) uses Q^i_{WR,s} which is not introduced at that point and uses indices i and s that are also not introduced.

• In equation (2) Q^i_{WR,r} is introduced but it is a different variable than Q^i_{WR,s}. This is not recommended because it uses the same variable for two different flows.

• The variables in the text have a different font then in the equations (for example compare equation (3) with line 130 and 131)

• Line 143: The meaning of variable X needs far more explanation. What is it representing? The same holds for X_r and X_{r_c}. I also recommend to always write into the sum over which indexes the sum is taken.

• In equations (4) new variables X_r^i appear that are also not introduced.

The readability would also improve significantly with the addition of a variable description table.

In the section Model parameterisation and validation

• Maybe use M^i_WR instead of M_WR in line 167 because the value per week is given.

• I think the readability would also improve quite a bit if you also write the variable name for the parameters, as you did in line 167.

• The addition, a table showing each parameter a value and a source would also significantly improve readability for the readers.

In figure 1, there is no “EE” node but an “EP” node. Is this an error?

The main function in the code currently has more than 2000 lines and therefore computations mentioned in the paper are hard to find. The code can be improved significantly if some parts inside the main function can be exported in different functions in other files.

I think the paper would also benefit from a further discussion of why the specific regions were chosen to reflect the continents. Large regions like Asia likely vary significantly between, e.g., West to East. It might therefore be necessary to further split such regions, while Oceania, for examply, seems rather insignificant compared to the other regions.

7. PLOS authors have the option to publish the peer review history of their article (what does this mean?). If published, this will include your full peer review and any attached files.

Reviewer #1: No

Reviewer #2: No

Reviewer #3: No

---

## [Author Response · Author response to Decision Letter 1]

13 Jan 2022

Reviewer 1

This is a well-written paper addressing a topical subject of the WEF nexus. The authors have provided every detail including the modelling code. The authors need to add references to some of the statements in the introduction and check the grammar. But otherwise this is a very good study.

Response:

Thank you for your valuable review comment. To address the concerns, we have altered the introduction to incorporate missing references and have carried out another grammar check. Relevant text is reproduced here:

line 5

Energy is an important contributor to human development and well-being, and it is also a key input to agriculture [1].

line 12

Additionally, excessive water withdrawal due to cheap energy may further deplete groundwater resources [2].

line 48

Currently, prima facie, the available water is adequate to match human consumption on an aggregate level [3]. 

line 64

Lastly, the SDG 6 measures are expected to improve the overall water management [4].

Other significant changes

To address comments of Reviewer 3, model description equations have been revised and tables summarizing parameter values and variable description are included.

Reviewer 2

This paper uses and expands the global “integrated cross-disciplinary model” known as Global Sustainability Model (GGSM) to investigate water sustainability under different future scenario of water demand. The authors claim that this is done while considering the food-water-energy nexus. The development of models of the food-water-energy nexus is a hot topic with potentially significant societal impacts. That said, I believe that the authors have not provided enough details to understand what exactly GGSM simulates and how. For example:

How are the different water resources allocated within each country?

Response:

We thank the reviewer for the comment.

We would like to clarify that the modeling of water resources within each country is not carried out in the current study. The aim of this work is to investigate the long-term regional demand-supply dynamics for various sectors and hence, analyzing the continent-level water stress (line 55). The

GGSM is a global model and hence, in order to explore the regional effect of water stress, the global water system is disaggregated into six continents. The allocation of water resources in these continents is modelled based on FAO, UN data [5] (line 243). However, internal allocation within the continent is not modelled and hence, modelling intra-national allocation of water resources is beyond the scope of the current study.

How are “economic and technological considerations” taken into account to assess how much water can be harvested?

Response:

The “economic and technological considerations” are taken into account by modelling the water reservoir in terms of exploitable water resources. Exploitable water is defined by AQUASTAT as follows (line 105 in the manuscript):

“Exploitable water resources (also called manageable water resources or water development potential) are considered to be available for development, taking into consideration factors such as: the economic and environmental feasibility of storing floodwater behind dams, extracting groundwater, the physical possibility of storing water that naturally flows out to the sea, and minimum flow requirements (navigation, environmental services, aquatic life, etc). Methods to assess exploitable water resources vary from country to country.”

How is water demand of the different sector satisfied?

Response:

The water demand by different sectors is satisfied by allocating available water to these sectors. The focus of the current work is modeling global water system, its linkages with GGSM, modelling demand and computing stress. The specific sequence of calculations is as follows. 

In order to elucidate the workings of the water system, a schematic diagram is

incorporated (main manuscript line 182). The diagram and associated description are

reproduced here: 

A schematic representation of the working of the water model of the GGSM is shown in Figure 1. On a global level, state variables are used with the sectoral intensities to obtain the total global sectoral demands. Geographical distributions for each of the sectors is used to disaggregate the total sectoral water demand into regional sectoral water demand. At this stage water demands for all the sectors and regions are available. Now, for each region all the sectoral demands are aggregated to get total regional water demand. Then, using the water availability for that region, regional water stress can be computed.

An example describing the model functioning is included in the main manuscript (line

190) and reproduced below:

The working of the water model can be understood through the following example:

For the industrial sector, industrial production would be used to obtain the global industrial water demand. Using geographical distribution, regional IS demand is obtained. For a region, for example, Africa, total regional demand is computed from regional demands for agriculture, energy, municipal and industrial demands. This total regional demand is used to compute the water stress for Africa. In the context of supplying these demands, the water reservoirs which represent the stock of exploitable water are utilized. Here, specific modes of demand-supply are not considered, and the competition between sectors is ignored. A simple water balance based on material flow analysis is employed. The water stress is computed for all regions and sectors. If the water stress is > 100%, it represents a situation where non-renewable water would be used.

How was the model validated (Figs. 2 and 3 are not clear at all)?

Response:

We would like to respond to this comment by first explaining how the water demand is determined and then providing details regarding validation.

Sectoral intensity trends represent the transformation functions to obtain the water demand for a particular sector from corresponding state variables. The data source and the computation method along with the example of use of these trends is incorporated in the Model Parameterization and Validation section (line 211) and reproduced below:

The sectoral intensity trends are used to compute the water demand for a sector using GGSM variable values. In other words, they represent the transformation functions to obtain the water demand for a particular sector from corresponding state variables. The historical sector-wise water demand data is obtained from the AQUASTAT database [8]. This country-level data is then aggregated continent-wise and mapped against state variable data (Y i ) for the same period to obtain the sectoral intensity plots (Ψ).

Now, the global sectoral demand model is validated by comparing the model outcomes (water demand for each of the sectors) with the historical data, as shown in Figure 4 (erstwhile Figure 3). The modified description of the validation (line 235) is reproduced below:

Figure 4 shows the validation of the global sectoral water demand computation model. The markers indicate the historical global water demand for agriculture, industry, and municipal sectors. The historical data of industrial sector water demand also includes the energy sector. Accordingly, the model outcomes’ global water demand of the energy and industrial sectors are aggregated before the comparison. The global water demand computed by the model for each sector, agriculture, municipal, and industrial (also including energy), conform well with the historical data.

What are the model limitations and assumptions?

Response:

Critical assumptions, simplifications and limitations are listed below:

Assumptions

1. Seawater is ignored.

2. Only exploitable water, that is, the total surface water and regular renewable groundwater, is considered.

3. Fossil groundwater, desalinated water, environmental water requirements, and flows are not considered.

4. Effect of climate change and extreme weather events are not modeled in the present effort, but they could be addressed using this paradigm in future work.

5. Economics of water supply-demand is not considered.

6. Historical sectoral water intensity trends are assumed to be valid in future and the possibility of a disruptive technology becoming available is ignored.

Limitations

This work was subject to several assumptions and simplifications, which also form a basis for potential avenues for future research. These are as follows:

1. This work focuses only on the demand side. It does not delve into detailed modeling of the recycling of water which is a critical factor in the context of high water stress region. Moreover, often the water use efficiency improvement is at the expense of energy resources. Hence, the dynamics between water recycling in water-stressed regions and energy requirements can shed light on a trade-off between efficiency improvement and water recycling.

2. Secondly, the economics of water is not considered. This is a crucial limitation. Incorporation of water pricing will permit modeling the impact of stress on water demand, thereby providing greater insights. 

3. The influence of climate change is another factor that is beyond the scope of the current work. Regional as well as global precipitation patterns can change because of climate change. Climate change may also influence the crop patterns, thus, altering the water intensity of the agricultural sector. Hence, one of the potential future research avenues could be the modeling the climate change influences on water demand and availability.

4. Lastly, as a simplification, the regional aggregation of countries is carried out based on the continent in which that they are part of. However, a better approach would be grouping the countries based on the river basins they are sharing. Another simplification is aggregate modeling of the agricultural water demand. Incorporation of different crops and their water demand could make the model outcomes more realistic.

These are included in the Modelling global water system section (line 86) and

conclusion section on (line 466).

Are the two-way interactions between food, water and energy taken into account? And if yes, how? I am sorry for the short review, but without these details, I am unable to provide any additional feedback on the results. At this stage, I thus recommend its rejection.

Response:

Thank you for your valuable review comment. Two way interactions

between different sectors mentioned by the reviewer are not taken into account in the

present work. We are currently working on capturing these feedback effects and two way

interactions. The scope of this work is limited to

• Modelling and integrating global water system with GGSM

• Capturing sectoral and regional demand dynamics

• Analysing the regional stress

Other significant changes

To address comments of Reviewer 3, model description equations have been revised and tables summarizing parameter values and variable description are included.

Reviewer 3

The authors present an interesting approach to model the water stress on the continents. However, there are still some parts that make it very hard to understand and therefore unsuitable for publication in its present form. The section Modeling global water system would improve a lot from a more detailed presentation of the equations and variables. Some problems currently are Equation (1) uses Q iW R,s which is not introduced at that point and uses indices i and s that are also not introduced.

Response:

Thank you for your valuable review comment. Relevant indices are included (revised model description is included in response to next comment).

In equation (2) Q iW R,r is introduced but it is a different variable than Q iW R,s . This is not recommended because it uses the same variable for two different flows.

Response:

The variable description has been updated as per the reviewer's suggestion; and , and are revised as , , and , respectively. Additionally, equations for regional water availability and water stress computation are included for clarity. Relevant text and equations (line 132) are reproduced below:

Equation 1 shows the utility of sectoral water intensity Ψ to obtain the total sectoral water demand T s for sector s at timestep i from GGSM variables Y. Here, s is a sector in Agriculture (P 1), Municipal (HH), Industry (IS), Energy (EP ), or Livestock (H1).

 1

 Equation 2 is used to compute the regional sectoral water demand Rs for region r and sector s.

 2

Total regional water demand T r can be computed by aggregating the regional sectoral water demand Rs for all sectors for a particular region as shown in Equation 3.

 3

 To compute the regional water stress, regional water availability $A$ is required. It can be computed using Equation \\ref{eq: regional availability}.

 4

 where, $M_{WR}^i$ represents global water availability at timestep $i$ and $Ga_{r}$ represents geographical distribution factor for availability for region $r$ at timestep $i$

 Now, using outcomes of Equations \\ref{eq: total regional demand} and \\ref{eq: regional availability}, regional water stress $Ws$ can be computed using Equation \\ref{eq: regional water stress}.

 5

The variables in the text have a different font then in the equations (for example compare equation (3) with line 130 and 131)

Response:

The model description is modified to ensure that variable text have same font in the equation and text. The same can be seen in the response of earlier comment.

Line 143: The meaning of variable X needs far more explanation. What is it representing? The same holds for and . I also recommend to always write into the sum over which indexes the sum is taken. 

Response:

Thank you for your comment. A more detailed description regarding variable has been included in the revised manuscript. Further, the section describing model has been modified to include an explicit and more readable (bulleted) description of the terms used in the equations. 

The variable over which sum is taken are mentioned in the equations. However, since these variables are summed over a list of entities (for example, countries or sectors), instead of including an index, the set of all the corresponding entities is incorporated in the description of the equation.

Relevant text and equations (line 155) are reproduced below:

The total value of the representative variable for region represented by can be obtained using Equation 6a. Its distribution, that is, contribution of region to the global value of sector is represented by . Here, is a representative variable in Agricultural area, GDP, Meat production, or Population; is a country in the list of countries in the world; and is a region in Africa, Asia, Europe, North America, Oceania or South America. Equation 6b shows computation of .

 6a

 6b

In equations (4) new variables X r i appear that are also not introduced.

Response:

To address the comment, new variables appearing in the equations have been introduced. A detailed description of modified equations involving X has been included in the response of the earlier comment.

The readability would also improve significantly with the addition of a variable description table. 

Response:

We thank the reviewer for the valuable review comment. A variable description table is included and the same is reproduced here (Table 1):

GGSM variable/compartment

Symbol

Carnivores

C1, C2

Energy production

EP

Fuel source

FS

Herbivores

Livestock

H1

Feral

H2, H3

Inaccessible resource pool

IRP

Industrial sector

IS

Inaccessible water reservoir

IWR

Primary producers

Agriculture

P1

Grasslands

P2

Forests

P3

Resource pool

RP

Water reservoir

WR

Water modelling variables

Availability of exploitable water (Regional)

Geographical distribution of water demand

Geographical distribution of water availability

Availability of exploitable water (Global)

Regional Sectoral water demand

Rs

Total Regional water demand

Tr

Total Sectoral water demand

Ts

Water Stress

Ws

Representative variable

X

State variable

Y

Sectoral Intensity

Subscripts

Country

c

Region in Africa, Asia, Europe, North America,Oceania, South America

r

Sector in Agricultural, Livestock, Industrial, Energy, Municipal

s

Representative variable in Agricultural area, Meat production, GDP, Population

v

Superscripts

Timestep

i

In the section Model parameterisation and validation Maybe use instead of in line 167 because the value per week is given

Response:

Thank you for your valuable review comment. and are are updated to and . Relevant text (line 205) is reproduced below:

Based on data from AQUASTAT [8], the total global exploitable water () and the total renewable water resources are about 135 and 1060 billion cu m per week, respectively. These quantities are assumed to be constant over the simulation horizon for the current work. Inaccessible water is the difference between total renewable water and total exploitable water. Hence, inaccessible water resources () are 925 billion cu m per week.

I think the readability would also improve quite a bit if you also write the variable name for the parameters, as you did in line 167.

Response:

To address the comment, the variable names are included in the parameterization section. Relevant portion of the revised text (line 212) is reproduced below:

The sectoral intensity trends are used to compute the water demand for a sector using GGSM variable values. In other words, they represent the transformation functions to obtain the water demand for a particular sector from corresponding state variables. The historical sector-wise water demand data is obtained from the AQUASTAT database [8]. This country-level data is then aggregated continent-wise and mapped against state variable data () for the same period to obtain the sectoral intensity plots (). Historical data is explicitly available for agricultural (), and municipal () sectors; however, separate historical data for the industrial () and energy () sectors are not available as the water withdrawal data for the industrial sector also includes the energy sector. Hoekstra [6] analyzed the global water consumption by various sectors. From 1996 to 2005, they computed the combined industry and energy sector water demand to be 400 billion cu m per year. Mekonnen et al [7] have computed annual global energy water for the period 2000-2005 to be around 250 billion cu m. Using this information, 62.5\\% of the aggregate industrial water demand can be allocated to the energy sector and 37.5\\% to the industrial sector. It is assumed that this allocation remains constant over the simulation horizon. The sectoral intensity for the livestock () sector is linearly proportional to its mass. 

The addition, a table showing each parameter a value and a source would also significantly improve readability for the readers.

Response:

Thank you for the comment. Parameter value and source depicting tables are now included. Further, to improve readability, the section addressing parameterization is split into two parts: first global sectoral demand parameterization and second regional sectoral demand parameterization. The parameterization summary is reproduced in Table 2 and Table 3:

Table 2. Summary of model parameterization

Parameter details

Values

Source

Historical water demand

and availability

M W R : 135 billion cu m/week

MIWR : 925 billion cu m/week

Agricultural, industrial and

municipal water withdrawal

AQUASTAT [8]

State variable values

Mass of P1, H1;

Production of IS, EE;

Population

Nisal et al [10]

IS/EP split of industrial water demand

37.5%/62.5%

Hoestra [6]

Mekonnen et al [7]

Regional availability distribution

Africa: 9%,

Asia: 28.4%,

Europe: 15.2%,

North America: 17%,

Oceania: 2.1%, and

South America: 28.3%

FAO,

United Nations [5]

Table 3. Representative variables for water demand from different sectors

Sector

Representative variable

Source

Agriculture

Agricultural area

11

Livestock

Meat production

12

Industry and Energy

GDP

13

Municipal

Population

14

In figure 1, there is no “EE” node but an “EP” node. Is this an error?

Response:

 Thank you for your valuable review comment. The text is corrected by replacing with .

The main function in the code currently has more than 2000 lines and therefore computations mentioned in the paper are hard to find. The code can be improved significantly if some parts inside the main function can be exported in different functions in other files.

I think the paper would also benefit from a further discussion of why the specific regions were chosen to reflect the continents. Large regions like Asia likely vary significantly between, e.g., West to East. It might therefore be necessary to further split such regions, while Oceania, for examply, seems rather insignificant compared to the other regions.

Response:

We thank the reviewer for the valuable comment. As suggested demand computation has been taken out of the main code.

Selection of the regions based on continents was a decision governed by challenges of the data discrepancy, that arose out of changing boundaries of the nations within a particular region. As a consequence of selecting continents as regions the future projections are immune to boundary changes. 

Further, we have acknowledged the limitation pertaining to this simplification in the conclusion section. It is also an avenue of research planned to be carried out in future. The relevant text (line 483) is reproduced below:

Lastly, as a simplification, the regional aggregation of countries is carried out based on the continent in which that they are part of. However, a better approach would be grouping the countries based on the river basins they are sharing. Another simplification is aggregate modeling of the agricultural water demand. Incorporation of different crops and their water demand could make the model outcomes more realistic. 

References

1. IEA, Paris Global Energy Review 2021; 2021. https://www.iea.org/reports/global-energy-review-2021..

2. Rodell, Matthew, Isabella Velicogna, and James S. Famiglietti. Satellite-based estimates of groundwater depletion in India. Nature 460.7258 (2009): 999-1002.

3. United Nations, The United Nations World Water Development Report 2021: Valuing Water. UNESCO, Paris https://unesdoc.unesco.org/ark:/48223/pf0000375724.

4. Sadoff, Claudia W., Edoardo Borgomeo, and Stefan Uhlenbrook. Rethinking water for SDG 6. Nature Sustainability 3.5 (2020): 346-347.

5. FOOD AND AGRICULTURE ORGANIZATION OF THE UNITED NATIONS. Review of World Water Resources by Country; 2003. http://www.fao.org/3/Y4473E/y4473e0f.gif.

6. Hoekstra AY. The water footprint of industry. In: Assessing and measuring environmental impact and sustainability. Elsevier; 2015. p. 221–254.

7. Mekonnen MM, Gerbens-Leenes P, Hoekstra AY. The consumptive water footprint of electricity and heat: a global assessment. Environmental Science: Water Research & Technology. 2015;1(3):285–297.

8. Food and Agriculture Organization of the United Nations (FAO). AQUASTAT Database; 2021. http://www.fao.org/aquastat/statistics/query/index.html?lang=en.

9. Jekel CF, Venter G. pwlf: A Python Library for Fitting 1D Continuous Piecewise Linear Functions; 2019. Available from: https://github.com/cjekel/piecewise_linear_fit_py.

10. Nisal A, Diwekar U, Hanumante N, Shastri Y, Cabezas H, Boumans R. Integrated model for food-energy-water (FEW) nexus to study global sustainability: The main generalized global sustainability model (GGSM). submitted to PLoS ONE. 2021.

11. Ritchie H, Roser M. Land Use. Our World in Data. 2013;.

12. Ritchie H, Roser M. Meat and Dairy Production. Our World in Data. 2017;.

13. Roser M. Economic Growth. Our World in Data. 2013;.

14. Roser M, Ritchie H, Ortiz-Ospina E. World Population Growth. Our World in

Data. 2013;.

---

## [Decision Letter · Decision Letter 2]

2 Mar 2022

PONE-D-21-28130R2Integrated model for Food-Energy-Water (FEW) nexus to

study global sustainability: The water compartments

and water stress analysisPLOS ONE

Dear Dr. Urmila,

Thank you for submitting your manuscript to PLOS ONE. After careful consideration, we feel that it has merit but does not fully meet PLOS ONE’s publication criteria as it currently stands. Therefore, we invite you to submit a revised version of the manuscript. The reviewers now recommend to accept your work for publication, however, you have archived your code in GitHub. GitHub is not a suitable repository. GitHub itself instructs authors to use other alternatives for long-term archival and publishing, such as Zenodo. Therefore, please, publish your code in one of the appropriate repositories according to our code and data policy. In this way, you must include in a potential reviewed version of your manuscript the new repository and its DOI.

We look forward to receiving your revised manuscript.

Kind regards,

Juan A. Añel

Academic Editor

PLOS ONE

Journal Requirements:

Reviewers' comments:

Reviewer's Responses to Questions

**Comments to the Author**

1. If the authors have adequately addressed your comments raised in a previous round of review and you feel that this manuscript is now acceptable for publication, you may indicate that here to bypass the “Comments to the Author” section, enter your conflict of interest statement in the “Confidential to Editor” section, and submit your "Accept" recommendation.

Reviewer #1: All comments have been addressed

Reviewer #3: All comments have been addressed

2. Is the manuscript technically sound, and do the data support the conclusions?

Reviewer #1: Yes

Reviewer #3: Yes

3. Has the statistical analysis been performed appropriately and rigorously? 

Reviewer #1: Yes

Reviewer #3: Yes

4. Have the authors made all data underlying the findings in their manuscript fully available?

Reviewer #1: Yes

Reviewer #3: Yes

5. Is the manuscript presented in an intelligible fashion and written in standard English?

Reviewer #1: Yes

Reviewer #3: Yes

6. Review Comments to the Author

Reviewer #1: The authors addressed all my review comments. This is a good and interesting manuscript. I recommend that it be accepted

Reviewer #3: (No Response)

7. PLOS authors have the option to publish the peer review history of their article (what does this mean?). If published, this will include your full peer review and any attached files.

Reviewer #1: No

Reviewer #3: No

---

## [Author Response · Author response to Decision Letter 2]

7 Mar 2022

Editor’s Comment

Comment: Thank you for submitting your manuscript to PLOS ONE. After careful consideration, we feel that it has merit but does not fully meet PLOS ONE’s publication criteria as it currently stands. Therefore, we invite you to submit a revised version of the manuscript. The reviewers now recommend to accept your work for publication, however, you have archived your code in GitHub. GitHub is not a suitable repository. GitHub itself instructs authors to use other alternatives for long-term archival and publishing, such as Zenodo. Therefore, please, publish your code in one of the appropriate repositories according to our code and data policy. In this way, you must include in a potential reviewed version of your manuscript the new repository and its DOI.

Response: We have put the code on Zenodo and new link is provided in the revised manuscript.

---

## [Editor Report · Decision Letter 3]

23 Mar 2022

Integrated model for Food-Energy-Water (FEW) nexus to

study global sustainability: The water compartments

and water stress analysis

PONE-D-21-28130R3

Dear Dr. Urmila,

We’re pleased to inform you that your manuscript has been judged scientifically suitable for publication and will be formally accepted for publication once it meets all outstanding technical requirements.

Kind regards,

Juan A. Añel

Section Editor

PLOS ONE
---

## [Editor Report · Acceptance letter]

25 Apr 2022

PONE-D-21-28130R3 

Integrated model for Food-Energy-Water (FEW) nexus to study global sustainability: The water compartments and water stress analysis 

Dear Dr. Diwekar:

I'm pleased to inform you that your manuscript has been deemed suitable for publication in PLOS ONE. Congratulations! Your manuscript is now with our production department. 

Kind regards, 

on behalf of

Dr. Juan A. Añel 

Section Editor

PLOS ONE